# Feeding ecology and activity patterns of *Hippopotamus amphibious* in the Dhidhessa Wildlife Sanctuary, Southwestern Ethiopia

Girma Gizachew Tefera[ID][1]*, Tadesse Habtamu Tessema[2], Tibebu Alemu Bekere[1], Tariku Mekonnen Gutema[1]

**1** Department of Natural Resource Management, Colleague of Agriculture and Veterinary Medicine, Jimma University, Jimma, Ethiopia, **2** Department of Biology, College of Computational Sciences, Jimma University, Jimma, Ethiopia

* kennaa20047@gmail.com

## Abstract

Understanding the dietary composition and activity patterns of hippopotamuses (Hippopotamus amphibious) is critical for assessing their ecological role within their habitat. This study investigated the feeding habits and behavioral rhythms of common hippopotamuses in the Dhidhessa Wildlife Sanctuary (DWS), Ethiopia, from 2022 to 2023. Dietary data were collected through direct observation of fresh feeding signs and fecal analysis, while activity patterns were recorded via continuous focal sampling of adult males and females at 30-minute intervals. Hippopotamuses allocated 30.1% of their time to resting, followed by movement (23.7%). Vocalizations (barking) constituted 53.1% of recorded behaviors, while yawning accounted for 46.9%. Males exhibited significantly more resting behavior than females. Peak feeding and movement occurred during early morning and late afternoon, whereas resting peaked at midday. Both sexes displayed higher frequencies of barking and yawning in the afternoon. Dietary analysis identified 34 plant species from 12 families consumed by hippos. Poaceae dominated their diet (60%), while Combretaceae contributed the least (0.5%). Typha latifolia was the most frequently consumed species (9.4%), followed by Eriochloa fatmensis (8.7%). Seasonal variation was evident, with 76.7% of foraging occurring in the wet season compared to 23.3% in the dry season. Twenty-one forage species were available year-round. These findings highlight the importance of seasonal resource availability and temporal activity shifts in hippopotamus ecology. Conservation strategies in DWS should prioritize habitat management to ensure sustainable ecosystem functionality.

## Introduction

### Background of the study

The common hippopotamus (*Hippopotamus amphibious*, hereafter referred to as "hippopotamus" or "hippo") was historically widespread across aquatic habitats

**Data availability statement:** Data is publicly accessible at: DOI: https://doi.org/10.6084/m9.figshare.28500605 Direct link: https://figshare.com/s/9463661c20b40348cc1f

**Funding:** The author(s) received no specific funding for this work.

**Competing interests:** The authors have declared that no competing interests exist.

throughout Africa [1], particularly in the lakes and rivers of sub-Saharan Africa [2]. In terms of body mass, it is the third-largest extant terrestrial mammal, surpassed only by the white rhinoceros (*Ceratotherium simum*) and elephants (*Loxodonta spp.*) [3]. The hippopotamus exhibits distinct morphological adaptations, including a large, related flattened cranium, short and robust limbs, a broad muzzle, and elongated tusk-like canines. Its sensory organs; eyes, ears, and nostrils are positioned high on the skull, facilitating respiration and environmental awareness while submerged [4]. A critical physiological constraint is its thin epidermis, which is highly susceptible to desiccation. To mitigate this, hippopotamuses secrete a viscous, reddish-brown cutaneous secretion, often referred to as "blood sweat," which functions as a natural sunscreen and antimicrobial agent [5]. However, due to the absence of true sweat glands, prolonged aerial exposure can lead to severe dermal cracking and dehydration, necessitating near-constant submersion during daylight hours.

The hippopotamus (*Hippopotamus amphibious*) is an obligate herbivore, exhibiting a strong dietary preference for graminoids, particularly short grasses [6,7]. As a fundamental niche dimension, diet plays a critical role in shaping animal evolution, life history strategies, and ecological function [8]). Consequently, understanding dietary composition is essential for ecological studies, as it provides key insights into species-environment interactions [9,10]. Such interactions influence broader ecological dynamics, including community structure, species diversity, relative abundance, and mechanisms of resource partitioning among sympatric species and conspecifics.

For the common hippopotamus (*Hippopotamus amphibious*), vegetation availability represents a critical limiting resource, second only to water in ecological importance [11,12]. As a semi aquatic mega herbivore, hippopotamuses exhibit unique ecological requirements, necessitating both aquatic habitat and terrestrial grazing areas [5]. Their crepuscular foraging behavior follows a predictable pattern: individuals depart aquatic habitats at dusk, traversing well-established trails spanning 3–5 km (with maximum recorded distances of 10 km) to access grazing areas [6]. Foraging bouts typically last up to five hours, with individuals returning to aquatic habitat before dawn [12,13]. This behavioral pattern imposes significant temporal and spatial constraints on their foraging ecology, distinguishing hippopotamuses from other African mega herbivores.

As selective grazers, hippos' individuals consume 35–50 kg of grass daily, representing approximately 2.5% of their body mass [12,13]. Despite their ecological significance, the species' dietary preferences and behavioral patterns remain poorly documented across Ethiopia, with particular knowledge gaps existing for populations within the DWS. Emerging reports indicate escalating human hippopotamus conflicts in the study region; with crop raiding behavior increasingly reported in adjacent farmlands. These conflicts frequently result in physical confrontations, occasionally causing injuries to local farmers attempting to protect their agricultural yields [14]. Such anthropogenic pressures, coupled with potential seasonal variations in resource availability, may be driving adaptive changes in the species' foraging strategies and activity rhythms. This study was designed to address critical knowledge gaps by systematically investigating the composition and seasonal variation of hippopotamus

diets and temporal patterns of activity budgets, with the ultimate goal of generating essential baseline data to inform science-based management strategies that effectively balance hippopotamus conservation with sustainable agricultural practices in the DWS ecosystem.

## Materials and methods

### Ethical statement

The research proposal (provided in the supplementary file) was reviewed and approved by the Review Board at Jimma University (JU) College of Agriculture, Veterinary Medicine, and Medicine. The Office of the Vice President subsequently reauthorized and accepted the request for research and community service following the Review Board's approval. JU granted ethical clearance for the consent process related to data collection tools (Reference No: RGS/752/2021).

Written informed consent was obtained from the participant for both participation and publication of case details, in accordance with the PLOS consent form.

### Description of the study area

The study was conducted in the DWS and its surrounding areas in southwestern Ethiopia, spanning three administrative zones: Jimma, Buno Bedele, and East Wollega, all within the Oromia National Regional State. The total area is approximately 1300 km². It is situated 395 km West of Addis Ababa, along the Addis Ababa Nekemte-Bedele route. Geographically, the location is between 8°30'0" and 8°40'0" N latitude and 36°22'0" and 36°43'0" E longitude (Fig 1 our team used shape files provided by the Ethiopian Mapping Agency to produce the map. The agency's official website offers free and open to researchers to access these shape files ([https://africaopendata.org/dataset/ethiopia-shapefiles](https://africaopendata.org/dataset/ethiopia-shapefiles))). The altitude in

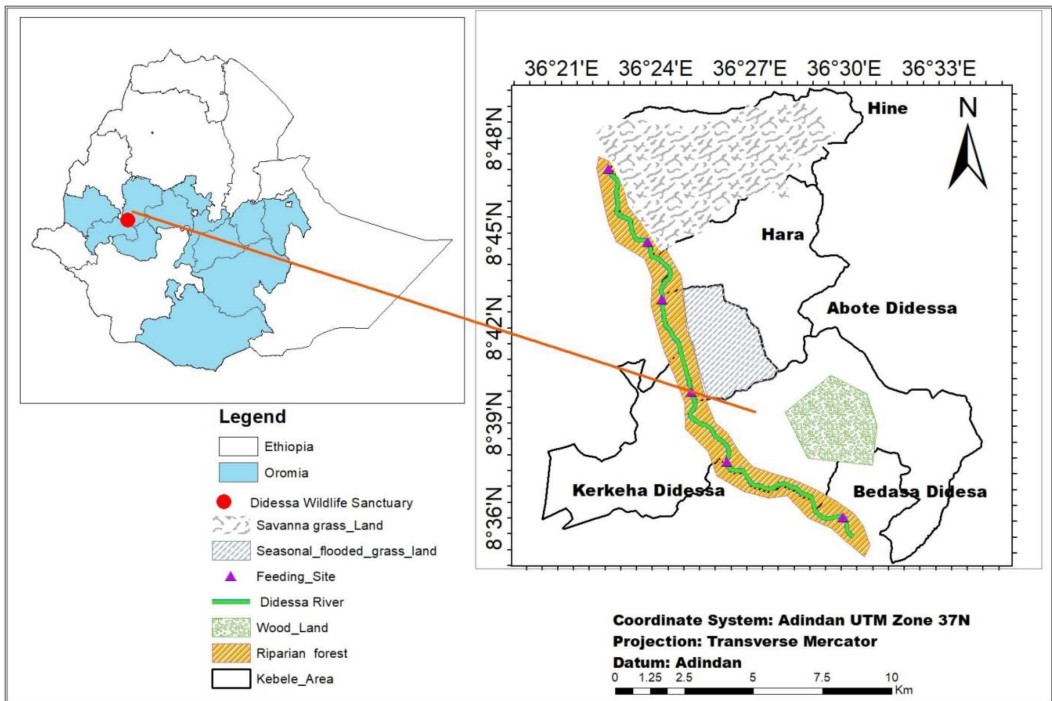

**Fig 1. Map of study area.** (Fig 1. our team used shape files provided by the Ethiopian Mapping Agency to produce the map. The agency's official website offers free and open to researchers to access these shape files ([https://africaopendata.org/dataset/ethiopia-shapefiles](https://africaopendata.org/dataset/ethiopia-shapefiles))).

the region varies from 1050 to 2560 meters. The area receives annual precipitation in a uni-modal pattern, ranging from 648 to 2001.8 mm. The Sanctuary has two distinct seasons: a dry season from November to February and a wet season from June to September. The lowland areas within the sanctuary tend to have warmer temperatures, with average minimum and maximum annual temperatures of 12°C and 35°C, respectively [15]. The study area is predominantly characterized by vegetation types such as *Combretum*, *Commiphora*, and *Acacia species*, interspersed with savanna grassland dominated by Hyparrhenia species.

## Methods

In order to record the activity patterns of hippos in the study area, a focal animal sampling method was used [16,17]. The selection of territorial adult males and adult females as focal sampling was based on their distinct ecological roles and representative behaviors within hippopotamus social systems, as this sex-based stratification captures: (1) behavioral plasticity, with males exhibiting agonistic, courtship, and resource defense behaviors while females demonstrate critical foraging strategies and social interactions essential for population persistence [6]; (2) demographic representativeness, since adults constitute >60% of typical populations and their activity budgets provide ecologically meaningful data for extrapolation [17,18]; (3) temporal consistency, as adults maintain stable behavioral patterns across seasons unlike sub adults undergoing behavioral maturation [17]; and (4) conflict relevance, given that territorial males are disproportionately involved in human-wildlife conflicts through wider nocturnal ranging [12] while females drive crop-raiding patterns due to nutritional demands [18]. This approach follows established ungulate sampling frameworks (Martin & Bateson, 2007) where ≥2 focal animals per sex provide >80% power to detect population-level behavioral patterns [5,6].

Behavioral data were collected using a stratified protocol combining direct observations and camera trap monitoring. Trained observers recorded hippopotamus behaviors at 30-minute intervals, with 5-minute inter-observation periods to minimize disturbance, while six motion-activated trail cameras deployed at key access points captured images at 5-minute intervals continuously. Data collection spanned six months across both dry (January-March) and wet (May-July) seasons, with sampling conducted five days per week (totaling 30 observation days per season) [17]. Behaviors were categorized into states (duration >30sec) including resting (stationary in water), standing (stationary on land), locomotion (walking/swimming), foraging (active grazing), and social interaction, as well as event behaviors (instantaneous) such as yawning (jaw extension >90°) and vocalization (barking displays) [16–18] (Tables S1 and S2 in S1 Appendix). In the study area, evidence of hippo feeding ecology was documented through direct observation of newly formed paths and trails. Researchers identified hippo tracks and feeding indicators, such as freshly grazed vegetation, to trace their foraging routes and determine the plant species consumed. By following fresh footprints and feeding signs along these pathways, plant species presumed to be grazed by hippos during nocturnal activity were recorded, as supported by previous studies [13,17,18] Whenever new hippo tracks were observed, vegetation assessments were conducted at feeding sites—specific locations where hippos grazed while traversing their foraging routes [18]. Plant specimens showing signs of hippo browsing, including bite marks and uprooting, were collected and taxonomically classified at the species level in the Botanical Herbarium of Wollega University for further analysis.

To complement feeding observations, fecal analysis was employed to identify plant species consumed by hippos, following established methodologies [19–21]. Fresh dung samples were collected across different habitats and air-dried before processing [22] (Figure S1 in S1 Appendix). Samples from the same season were pooled to form composite samples, reducing seasonal variability [23]. Each composite sample was manually disaggregated and washed with water through a 0.1 mm sieve to remove finely digested material [24]. The remaining particulate matter was transferred to a beaker, treated with 75% ethanol, and left for 48 hours. After decanting the ethanol, domestic bleach was added to facilitate tissue bleaching, which continued until complete digestion of non-cuticular matter [25,26]. The samples were then rinsed with distilled water, stained with safranin for enhanced visibility, and allowed to settle for 48 hours [25].

Microscopic analysis was conducted using a compound light microscope, with ten slides prepared per composite sample. A subsample of the processed fecal material was placed on a gridded slide and examined at ×40 and ×100 magnifications [27]. Epidermal fragments were identified by comparing them with reference slides [18] and assessing diagnostic features such as cell morphology, venation patterns, and trichome presence [28]. Identifiable fragments were classified into functional groups (dicots, herbs, and graminoids) [22,29], while unidentifiable remains were recorded as unknown.

## Data analysis

The relative feeding preference for each plant species was calculated by determining the proportion of its observed feeding signs relative to the total feeding occurrences recorded across all species. The formula used was:

$$CSi = \frac{FSi}{\sum_{i=1}^{n} FSt} x100$$

where:

- FSi = Frequency of feeding signs observed for species *i* (number of occurrences along transects within a plot),

- CSi = Contribution of species *i* to total feeding frequency (expressed as a percentage),

- $\sum_{i=1}^{n}$ FSt = Sum of feeding frequencies for all recorded species.

This metric reflects both the relative abundance and selectivity of hippos toward specific plant species within their foraging habitat. Additionally, behavioral differences between male and female hippos were assessed using independent-sample *t*-tests, with statistical analyses performed in Origin Pro 8.5.1. This approach allowed for the comparison of feeding patterns and movement behaviors between sexes, providing insights into potential sex-based foraging strategies.

Dietary composition percentages were calculated by assessing the proportion of plant material via frequency counts. This approach facilitated a comprehensive reconstruction of hippopotamus diet composition, incorporating both direct observation of feeding evidence and microscopic fecal remains.

## Results

### Diurnal activity and event patterns of hippopotamus

A comprehensive inventory was constructed from 4,480 behavioral records obtained through focal sampling of adult male and female hippopotamus. Of these records, 43.3% (n = 1,941) represented sustained activities, while 56.7% (n = 2,539) documented discrete behavioral events across thirty observation days. Quantitative analysis revealed a significant predominance of resting behavior, accounting for 30.1% of observed activity budgets (Table 1). In contrast, vocalizations (barking) and yawning displays represented 53.1% and 46.8% of recorded behavioral events respectively (Figure S2 in S1 Appendix), suggesting these behaviors constitute frequent components of hippopotamus behavioral repertoires.

### Time-specific activity budgets

Significant temporal variations were observed in the activity budgets of both male and female hippopotamuses throughout the diurnal cycle. Feeding, moving, and standing behaviors exhibited bimodal peaks, with heightened activity occurring

**Table 1. Diurnal activity and event records of male and female hippopotamus in the study area.**

| Sex | Behavioural activities | | | | | Total | Behavioural events | | Total |
|---|---|---|---|---|---|---|---|---|---|
| | Feeding | Resting | Moving | Standing | Socializing | | Barking | Yawning | |
| Male | 179 | 325 | 172 | 187 | 103 | 946 | 629 | 580 | 1209 |
| Female | 245 | 259 | 288 | 81 | 102 | 995 | 720 | 610 | 1330 |
| Total | 424 | 584 | 460 | 268 | 205 | 1941 | 1349 | 1190 | 2539 |
| % | 21.8 | 30.1 | 23.7 | 13.8 | 10.6 | 43.3 | 53.1 | 46.9 | 56.7 |

during crepuscular periods (07:00–09:00am and 16:00–18:00 pm) (Fig 2A and 2B). In contrast, recumbent resting behavior predominated during peak solar radiation hours (11:00am–15:00 pm), likely reflecting thermoregulatory avoidance strategies (Fig 2A and 2B).

Vocalizations (barking) and yawning displays were observed throughout the observation period, with increased frequency during afternoon hours. Yawning behavior was characterized by cephalic elevation accompanied by pronounced

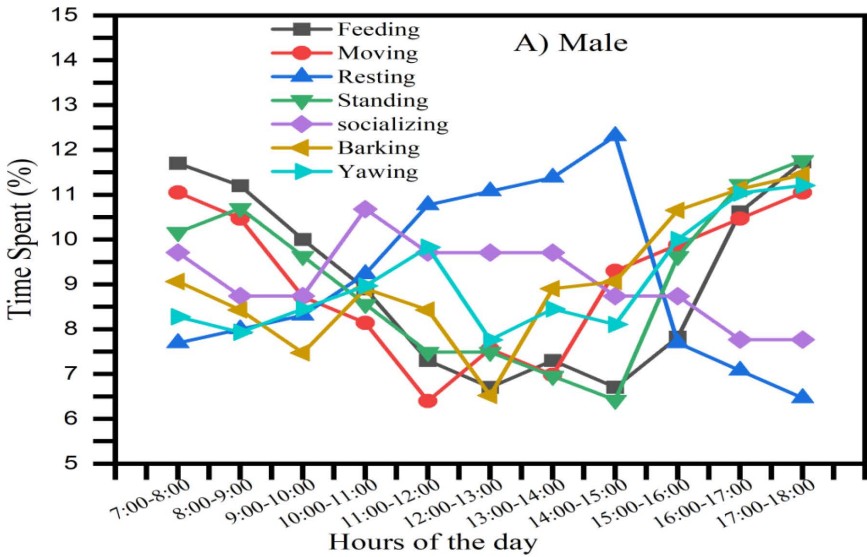

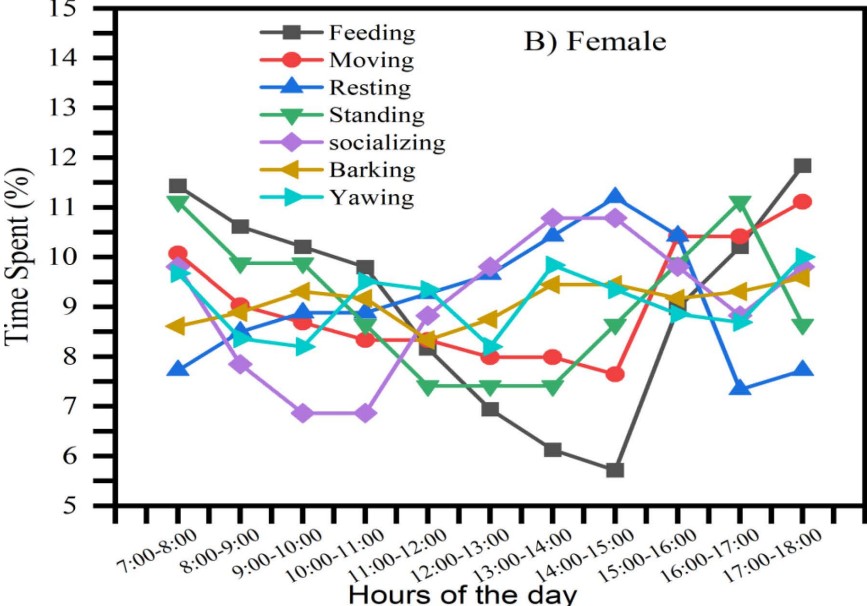

**Fig 2. Daily activity budgets (percentage of time) in the study area. (A)** Daily activity of male hippo. **(B)** Daily activity of female hippo.

jaw extension (Fig 3). Post-yawn observations revealed repetitive oral movements accompanied by low-frequency vocalizations and accentuated head movements. The yawning behaviors of male and female hippopotamuses were found to be quite similar.

## Monthly variations in activity and event budget

Analysis of monthly behavioral records revealed significant seasonal patterns in Hippopotamuses activity budgets (Table 2). Feeding frequency peaked in June, with secondary elevation in May, while minimal foraging activity was recorded during January and February. Movement patterns mirrored this trend, with maximal movement occurring in June and comparable activity levels observed in February and July.

Resting behavior demonstrated distinct seasonal variation, reaching maximal occurrence in March, followed closely by February. Conversely, January and July exhibited the lowest incidence of stationary behaviors (standing and moving, respectively). Social interactions, including mating behaviors, showed bimodal seasonality with peaks in February-March and notable reduction in July. Vocalization patterns followed similar seasonal trends, with barking frequency highest in March, followed by February, and minimal occurrences in July. Yawning displays exhibited parallel seasonality, with maximal observations in March and minimal frequency in July and May (Table 2).

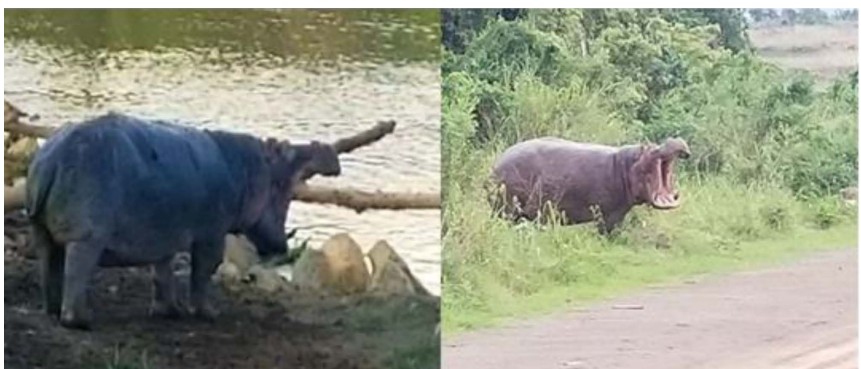

**Fig 3. Yawning characteristics of hippopotamus in the study area.**

**Table 2. Monthly variations of activities and events of male and female hippopotamus.**

| Months | Behavioural activities | | | | | | Behavioural events | | |
|---|---|---|---|---|---|---|---|---|---|
| | Feeding | Resting | Moving | Standing | Socializing | Total | Barking | Yawning | Total |
| January | 55 | 102 | 55 | 53 | 34 | 299 | 224 | 197 | 421 |
| February | 60 | 142 | 63 | 69 | 39 | 373 | 274 | 217 | 491 |
| March | 69 | 163 | 76 | 68 | 38 | 414 | 300 | 257 | 557 |
| May | 81 | 76 | 97 | 34 | 34 | 322 | 195 | 201 | 396 |
| June | 86 | 55 | 105 | 23 | 32 | 301 | 182 | 169 | 351 |
| July | 73 | 46 | 64 | 21 | 28 | 232 | 174 | 149 | 323 |
| Total | 424 | 584 | 460 | 268 | 205 | 1941 | 1349 | 1190 | 2539 |
| (%) | 21.8 | 30.1 | 23.7 | 13.8 | 10.6 | | 43.3 | 53.1 | 46.9 | 56.7 |

## Comparison in activity and event budget

Quantitative analysis of 4,480 behavioral records (1,941 activities and 2,539 events) revealed subtle sexual differences in Hippopotamuses time allocation (Table 1). Females accounted for 51.3% of observed activities and 52.4% of behavioral events, while males demonstrated marginally lower engagement (48.7% and 47.6% respectively).

The diurnal activity partitioning of hippopotamuses revealed sex-specific patterns, with males allocating 32.2% (Fig 4A), followed by standing (19.8%) (Refer to Figure S3 in S1 Appendix), while females exhibited greater mobility (28.9%) with comparable resting time (28%) and followed by feeding activity (24%) (Fig 4B). However, independent samples t-test analysis indicated no statistically significant sexual dimorphism in overall activity patterns (t=2.18, df=6, P>0.05), suggesting that despite these quantitative differences in time budgeting, the fundamental behavioral ecology remains consistent between sexes during daylight periods.

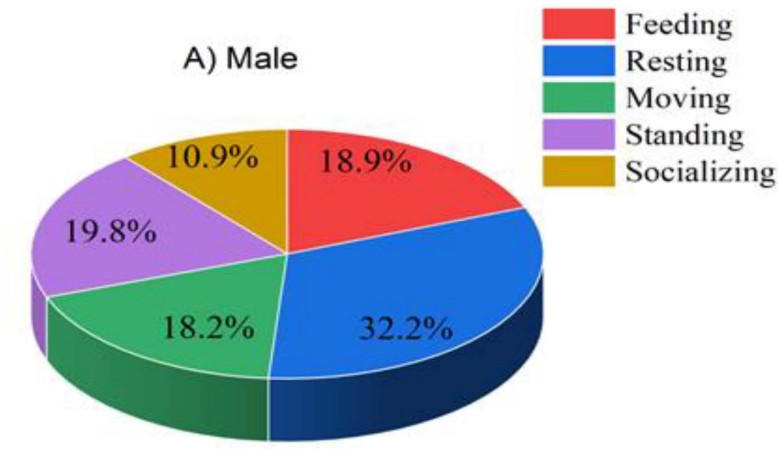

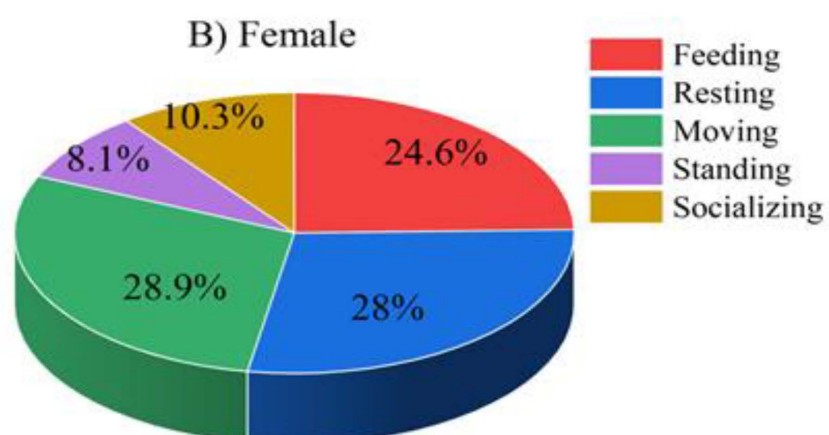

**Fig 4. Major behavioral activities of hippopotamuses categorized by sex. (A)** Major behavioral activities of male hippo. **(B)** Major behavioral activities of female hippo.

## Seasonal variations in activity budget

Analysis of seasonal activity budgets revealed distinct behavioral shifts between wet (n = 1110 activities, 1070 events) and dry seasons (n = 831 activities, 1469 events). During the dry season, hippopotamuses predominantly engaged in resting (37.1%), standing (17.8%), and vocalizations (barking: 54.3%), with these behaviors being significantly more prevalent than during wet periods. In contrast, the wet season was characterized by increased feeding (28.9%), moving (walking: 31.8%), and yawning displays (48.5%). Statistical analysis confirmed significant seasonal differences in activity budgets (t (6) = 2.44, p < 0.05) (Fig 5), suggesting that environmental conditions substantially influence behavioral allocation patterns. These findings demonstrate how seasonal resource availability and potentially thermoregulatory demands shape hippopotamus behavioral ecology.

## Feeding ecology

Feeding observation analysis was identified 34 plant species across 12 families in the hippopotamus diet (Table 3). The Poaceae family dominated dietary composition, representing 60% of total intake, while Combretaceae constituted only 0.5% of consumed vegetation (Figure S4 in S1 Appendix). At the species level, *Typha latifolia* emerged as the primary dietary component (9.4% of total intake), followed by *Eriochloa fatmensis* (8.7%). In contrast, *Oryza sativa* and *Chamaeurista mimosoides* each accounted for merely 0.3% of dietary content, with *Psoralea bituminosa* representing the least consumed species at 0.2%. These findings demonstrate significant selectivity in hippopotamus foraging behavior, with strong preference for graminoid species.

In addition to native vegetation documented within the DWS, the analysis revealed significant crop depredation by hippopotamuses in adjacent agricultural areas. Six cultivated species were regularly consumed, with the graminoid crops *Zea mays* (maize) and *Saccharum officinarum* (sugar cane) representing the most frequently targeted species. Other important agricultural components of the hippopotamus diet included *Musa paradisiaca* (banana), *Sorghum bicolor* (great millet), and *Allium sativum* (garlic) and *Allium cepa* (onion). These findings demonstrate the species' adaptive foraging behavior and its capacity to utilize anthropogenic food resources when available, potentially indicating habitat compression or nutritional supplementation strategies.

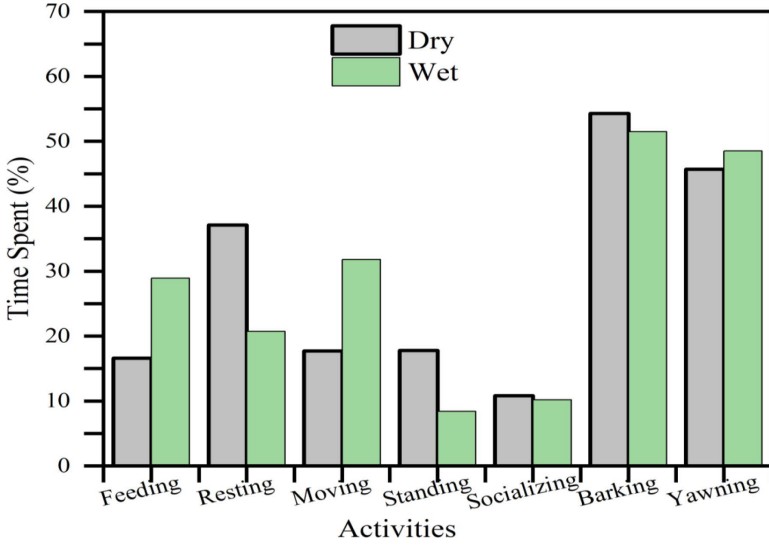

**Fig 5. Activity budgets of hippopotamus across seasons in study area.**

**Table 3. Plant species eaten by common hippopotamus in DWS.**

| Family | Species | Frequency of observation | Percentage |
|---|---|---|---|
| Poaceae | *Echinochloa pyramidalis* | 62 | 5.4 |
| Poaceae | *Coelorhachis afraurita* | 6 | 0.5 |
| Poaceae | *Eriochloa fatmensis* | 99 | 8.7 |
| Poaceae | *Cynodon dactylon* | 75 | 6.6 |
| Poaceae | *Cynodon plectostachyus* | 38 | 3.3 |
| Poaceae | *Sacciolepis africana* | 65 | 5.7 |
| Poaceae | *Cymbopogon commutatus* | 35 | 3.1 |
| Poaceae | *Hyparrhenia glabriuscula* | 30 | 2.6 |
| Poaceae | *Cynodon nlemfuensis* | 43 | 3.8 |
| Poaceae | *Cymbopogon citratus* | 37 | 3.3 |
| Poaceae | *Hyparrhenia rufa* | 19 | 1.7 |
| Poaceae | *Hyparrhenia hirta* | 27 | 2.4 |
| Poaceae | *Hyparrhenia cymbaria* | 7 | 0.6 |
| Poaceae | *Panicum maximum* | 5 | 0.4 |
| Poaceae | *Oryza sativa* | 3 | 0.3 |
| Poaceae | *Echinochloa cruspavonis* | 68 | 6 |
| Poaceae | *Leptochloa rupestris* | 56 | 4.9 |
| Cyperaceae | *Cyperus rigidifolius* | 83 | 7.3 |
| Fabaceae | *Vigna vexillata* | 5 | 0.4 |
| Fabaceae | *Chamaeurista mimosoides* | 3 | 0.3 |
| Fabaceae | *Psoralea bituminosa* | 2 | 0.2 |
| Fabaceae | *Laburum anagyroide* | 6 | 0.5 |
| Onagraceae | *Epilobium hirsutum* | 8 | 0.7 |
| Commelinaceae | *Commelina benghalensis* | 35 | 3.1 |
| Amaranthaceae | *Alternanthera nodiflora* | 67 | 5.9 |
| Asteraceae | *Guizotia scabra* | 8 | 0.7 |
| Asteraceae | *Galinsoga parviflora cov.* | 5 | 0.4 |
| Apiaceae | *Hydrocotyle mannii* | 42 | 3.7 |
| Combretaceae | *Combretum pancultum* | 6 | 0.5 |
| Lamiaceae | *Orthosiphon schimperi* | 31 | 2.7 |
| Polygonaceae | *Fallopia baldschanira* | 15 | 1.3 |
| Polygonaceae | *Persicaria decipiens* | 17 | 1.5 |
| Polygonaceae | *Persicaria attenuate* | 24 | 2.1 |
| Typhaceae | *Typha latifolia* | 107 | 9.4 |
| | Total | 1138 | 100 |

The analysis revealed significant seasonal differences in the dietary ecology of hippos ($t = 2.03$, $df = 33$, $p < 0.05$) (Table 4). During the wet season, hippos demonstrated greater dietary diversity, consuming 76.7% of available forage species compared to only 23.3% during the dry season. The wet season diet was dominated by *Typha latifolia* (9.9% of total intake) and *Eriochloa fatmensis* (9.2%), while *Chamaeurista mimosoides* and *Psoralea bituminosa* represented marginal components (0.2% each). In contrast, the dry season diet showed increased reliance on *Echinochloa pyramidalis* (15.1%), with *Typha latifolia* remaining important (7.9%) and *Chamaeurista mimosoides* (0.4%) being the least consumed species. Notably, 21 forage species were utilized year-round, demonstrating the species' dietary flexibility.

**Table 4. Plant species consumed by hippos during the wet and dry seasons in DWS.**

| Species | Wet season | Dry season | Total |
|---|---|---|---|
| *Echinochloa pyramidalis* | 20 (2.3%) | 40 (15.1%) | 62 (5.4%) |
| *Coelorhachis afraurita* | 4 (0.5%) | 2 (0.8%) | 6 (0.5% |
| *Eriochloa fatmensis* | 80 (9.2%) | 19 (7.2%) | 99 (8.7%) |
| *Cynodon dacylon* | 60 (6.9%) | 15 (5.7%) | 75 (6.6%) |
| *Cynodon plectostachyus* | 28 (3.2%) | 10 (3.8%) | 38 (3.3%) |
| *Sacciolepis africana* | 56 (6.4%) | 9 (3.4%) | 65 (5.7%) |
| *Cymbopogon commutatus* | 22 (2.5%) | 13 (4.9%) | 35 (3.1%) |
| *Hyparrhenia glabriuscula* | 30 (3.4%) | – | 30 (2.6%) |
| *Cynodon nlemfuensis* | 23 (2.6%) | 20 (7.5%) | 43 (3.8%) |
| *Cymbopogon citratus* | 21 (2.4%) | 17 (6.4%) | 37 (3.3%) |
| *Hyparrhenia rufa* | 14 (1.6%) | 5 (1.9%) | 19 (1.7%) |
| *Hyparrhenia hirta* | 20 (2.3%) | 7 (2.6%) | 27 (2.4%) |
| *Hyparrhenia cymbaria* | 5 (0.6%) | 2 (0.8%) | 7 (0.6%) |
| *Panicum maximum* | 5 (0.6%) | – | 5 (0.4%) |
| *Oryza sativa* | 3 (0.3%) | – | 3 (0.3%) |
| *Echinochloa cruspavonis* | 50 (5.7%) | 18 (6.8%) | 68 (6%) |
| *Leptochloa rupestris* | 45 (5.2%) | 11 (4.2%) | 56 (4.9%) |
| *Cyperus rigidifolius* | 65 (7.4%) | 18 (6.8%) | 83 (7.3%) |
| *Vigna vexillata* | 3 (0.3%) | 2 (0.8%) | 5 (0.4%) |
| *Chamaeurista mimosoides* | 2 (0.2%) | 1 (0.4%) | 3 (0.3%) |
| *Psoralea bituminosa* | 2 (0.2%) | – | 2 (0.2%) |
| *Laburum anagyroide* | 6 (0.7%) | – | 6 (0.5%) |
| *Epilobium hirsutum* | – | 8 (3%) | 8 (0.7%) |
| *Commelina benghalensis* | 30 (3.4%) | 5 (1.9%) | 35 (3.1%) |
| *Alternanthera nodiflora* | 67 (7.7%) | – | 67 (5.9%) |
| *Guizotia scabra* | 8 (0.9%) | – | 8 (0.7%) |
| *Galinsoga parviflora cov.* | 5 (0.6%) | – | 5 (0.4%) |
| *Hydrocotyle mannii* | 30 (3.4%) | 12 (4.5%) | 42 (3.7%) |
| *Combretum pancultum* | – | 6 (2.3%) | 6 (0.5%) |
| *Orthosiphon schimperi* | 31 (3.6%) | – | 31 (2.7%) |
| *Fallopia baldschanira* | 11 (1.3%) | 4 (1.5%) | 15 (1.3%) |
| *Persicaria decipiens* | 17 (1.9%) | – | 17 (1.5%) |
| *Persicaria attenuate* | 24 (2.7%) | – | 24 (2.1%) |
| *Typha latifolia* | 86 (9.9%) | 21 (7.9%) | 107 (9.4%) |
| Total | 873 (76.7%) | 265 (23.3%) | 1138 (100%) |

An analysis of 80 hippopotamus (*Hippopotamus amphibious*) fecal samples revealed significant seasonal variations in dietary composition, including dicots, graminoids, forbs (herbs), and unidentified plant material (Fig 6). A statistically significant difference was observed in graminoid consumption between wet and dry seasons (t = 1.95, df = 1, P < 0.05). Although dicot consumption was higher during the dry season compared to the wet season, this difference was not statistically significant (t = 5.4, df = 1, P > 0.05). In contrast, forb intake was significantly greater during the wet season (t = 2.76, df = 1, P < 0.05). These findings demonstrate a seasonal shift in dietary preferences, likely driven by fluctuations in plant availability across different seasons (S3 Table in S1 Appendix).

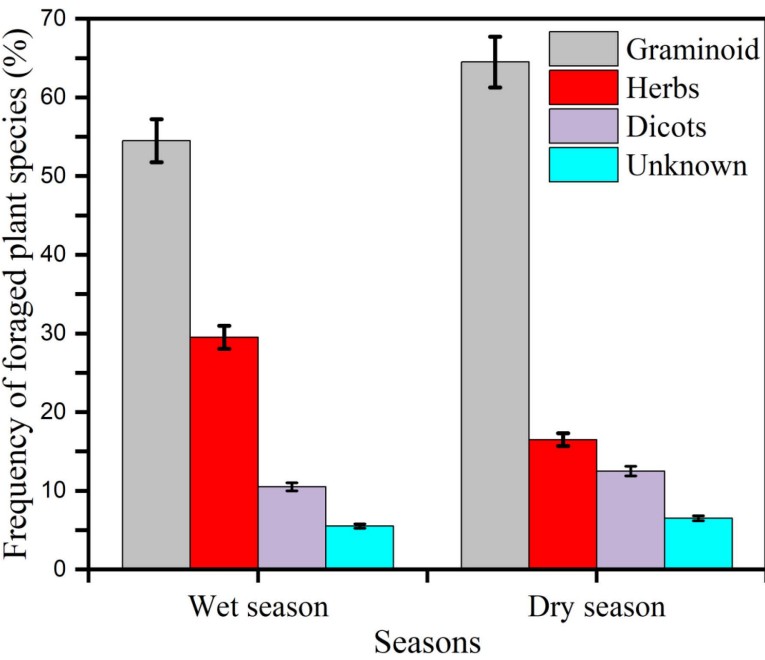

**Fig 6. Percentage occurrence of plant species in hippopotamus diet derived from faceal analysis in the DWS.**

## Discussion

### Activity pattern

Understanding the fundamental behaviors of hippopotamuses is critical for assessing population health, ecosystem dynamics, and the potential for sustainable coexistence with human communities [6,17]. The findings of this study revealed that hippos exhibited a variety of daytime behaviors, with resting being the most prevalent, followed by walking, feeding, standing, and social interactions. These results are consistent with previous research by [18] in Chebera Churchura National Park and [30] in Boye Wetland, Ethiopia. However, the current findings contrast with those of [17], who reported that hippos spent more than half of the day resting. This divergence may stem from methodological differences, as the present study employed focal sampling, whereas [17] utilized scan sampling. Additionally, variations were observed in specific behavioral events, such as barking and yawning, among individual hippopotamuses.

The findings of this study indicate that hippopotamuses exhibit a bimodal activity pattern, with peak activity occurring in the early morning and late afternoon, followed by a resting phase during midday. In the morning, individuals engaged in feeding bouts, frequently returning to the water, which appeared to induce intermittent restlessness. During the hottest hours of the day, the majority of observed hippopotamuses remained inactive, a behavioral adaptation that likely minimizes thermoregulatory stress and prevents cutaneous desiccation [31]. Feeding activity largely ceased around midday, except on days with precipitation. These observations align with previous reports by [17] as well as [30] and [18], who noted increased movement and aquatic retreats during periods of elevated temperatures. Notably, the present study found that hippopotamuses maintained partial submersion even while feeding, consistent with the findings of [6,18,30,32], who documented individuals submerging most of their bodies while extending their necks to graze on emergent or floating vegetation. The present study conducted in DWS revealed monthly variations in the activity budget of hippopotamuses, likely influenced by factors such as water availability, preferred vegetation proximity, and ambient temperature. Feeding activity peaked in June, followed by May, while the lowest levels were recorded in February and March [17,18]. During these

latter months, elevated temperatures—attributed to reduced rainfall and limited cloud cover—resulted in prolonged resting behavior, with individuals either fully submerged in water or seeking shade, consequently reducing their food intake. In contrast, cloudier conditions in June appeared to stimulate increased movement and foraging activity. These findings align with previous studies [17,31].

Daytime movement declined notably in July, potentially due to the abundance of tender grasses available near resting sites during nocturnal feeding bouts, coupled with decreased competition from livestock. This reduction in movement may also be linked to environmental constraints, such as extensive flooding and water overflow, as well as anthropogenic disturbances like agricultural activities, which restricted hippopotamus mobility. These findings align with previous studies [18,30,33], which reported that elevated water levels submerge grazing areas, thereby limiting foraging movements.

Additionally, a marked increase in barking and yawning was observed in March. This behavioral shift may be associated with rising temperatures and the seasonal burning of savanna grasslands. Similar patterns were documented by [6]), and [18], who noted heightened frequencies of these behaviors during March and April, possibly linked to mating-related interactions.

Empirical evidence suggests that sex-based behavioral differences in hippopotamuses are mediated by multiple ecological and social factors, including environmental disturbances, nutritional requirements, social play, agonistic interactions, and reproductive behaviors. The present study's findings demonstrate that female hippopotamuses exhibit significantly greater behavioral activity than their male counterparts. This observed sexual dimorphism in activity patterns may be attributed to hippopotamuses' selective preference for resting sites that optimize resource availability (particularly aquatic habitats with appropriate depth for full body submersion) while minimizing anthropogenic disturbances.

These results corroborate previous findings by [6], who reported that female hippopotamuses display heightened aggressive behaviors and vocalizations when protecting offspring from potential threats, including male conspecifics. Specifically, maternal females were frequently observed engaging in defensive behaviors such as barking and yawning as protective mechanisms for their calves. Similar observation were documented by [17; 18,30] in various hippopotamus populations.

Seasonal variations exert significant influence on the behavioral ecology of hippopotamuses. The present study's findings demonstrate distinct seasonal behavioral patterns, with dry season observations dominated by resting and standing behaviors, while rainy season observations showed increased frequencies of feeding and locomotion. Furthermore, agonistic behaviors (barking and yawning) exhibited pronounced seasonal variation, occurring with greater frequency during dry months, likely associated with both resource competition and reproductive activities. These behavioral shifts appear directly correlated with environmental parameters, particularly fluctuations in water levels and resource availability, as well as anthropogenic disturbances. The observed patterns align with previous research by [6,17,34], who similarly documented seasonal behavioral adaptations in hippopotamus populations. Specifically, the current study's results suggest that hippopotamuses modify their activity budgets in response to seasonal environmental stressors, with dry season conditions promoting energy conservation behaviors and wet season conditions facilitating increased foraging and movement.

### Feeding ecology

Current understanding of hippopotamus feeding ecology remains limited, with existing studies predominantly relying on qualitative rather than quantitative data [35]. The present dietary survey conducted in DWS and adjacent areas documented hippopotamuses consuming 34 distinct plant species across both wet and dry seasons. This finding corroborates research by [13] in Burkina Faso's Biosphere Reserve (34 species) while showing slightly lower diversity than [18] report of 40 species from Chebra Churchura National Park, Ethiopia. The higher species richness observed in Chebra Churchura likely reflects the park's superior biodiversity conservation status, minimal anthropogenic disturbance, and greater ecosystem heterogeneity.

Notably, the study identified greater dietary diversity than reported by [36] in Arjo Dhidhessa, Ethiopia; [30] in Boye Wetland, Ethiopia; and [12] in Zambia's Luangwa River. These discrepancies may be attributed to methodological differences

in study duration and/or anthropogenic impacts on local flora, particularly through agricultural expansion and settlement encroachment – a phenomenon previously documented by [13] and [37].

Analysis of dietary preferences revealed *Typha latifolia*, *Echinochloa fatmensis*, *Cyperus rigidifolius*, *Cynodon dactylon*, and *Sacciolepis africana* as dominant components of the hippopotamus diet. Of particular note, *Cynodon dactylon* emerges as a consistently important dietary item across multiple geographic regions, as evidenced by [12,18,36].

The study revealed significant seasonal variation in dietary composition, with wet season foraging encompassing 76.7% of identified plant species compared to only 23.3% during dry periods. While hippopotamuses accessed twenty-one forage species throughout both seasons, species diversity markedly declined as the dry season progressed. This temporal pattern in resource utilization appears directly correlated with phenological changes in grass and herbaceous plant availability within the ecosystem. These findings are consistent with previous research by [18] and [36].

Fecal analysis demonstrated that foraged species comprised three functional groups: graminoids, dicots, and herbs, with proportional representation varying significantly between seasons. Notably, grasses (Poaceae) and sedges (Cyperaceae) formed the dietary foundation across both seasonal periods. This dietary pattern confirms the species' classification as a bulk grazer while demonstrating behavioral plasticity in response to seasonal resource fluctuations – maintaining core grass consumption while opportunistically incorporating other plant functional types [18]. The predominance of graminoids in the observed diet aligns with established literature characterizing hippopotamuses as obligate grazers [5,6]. The current findings further corroborate regional studies by [18,36], confirming the ecological consistency of hippopotamus grazing behavior across diverse African ecosystems while demonstrating local adaptations to seasonal resource availability.

## Conclusion

This study elucidates the behavioral ecology and dietary adaptations of hippopotamuses in the DWS ecosystem of southwestern Ethiopia. The findings demonstrate that hippopotamuses exhibit significant behavioral plasticity in response to local ecological conditions, with distinct sex-specific activity patterns. Resting emerged as the predominant behavioral state for both sexes, though males allocated substantially more time to this activity than females. In contrast, females displayed greater overall behavioral activity, particularly in locomotion, vocalizations (barking), and yawning behaviors pattern likely reflecting their heightened parental investment and territorial defense requirements. The observed bimodal activity pattern, with peaks during crepuscular periods and midday resting phases, aligns with thermoregulatory adaptations to tropical climates. Furthermore, they documented significant temporal variation in behavioral patterns across seasonal and monthly cycles, reflecting adaptive responses to fluctuating environmental conditions.

Dietary analysis revealed the species' ecological flexibility as a bulk grazer, with graminoids constituting 66% of the diet while incorporating seasonal variations in herbaceous and shrub species consumption. The maintenance of core grass consumption (Poaceae and Cyperaceae families) throughout both wet and dry seasons, coupled with opportunistic incorporation of other functional plant groups, demonstrates nutritional adaptation to seasonal resource availability. These findings underscore DWS's ecological capacity to support viable hippopotamus populations, contingent upon: maintenance of critical aquatic and riparian habitats, protection of key grazing resources, and minimization of anthropogenic disturbances [38,39]. Finally, this study contributes to understanding hippopotamus behavioral ecology in understudied Ethiopian wetlands, providing critical baseline data for conservation planning in this ecologically significant region. Future research should incorporate longitudinal monitoring to assess population trends and habitat quality under changing climatic conditions and increasing anthropogenic pressures.

## Supporting information

**S1 Appendix. Supplementary.**
(DOCX)

## Acknowledgments

We thank to IDEA-WILD for their material support. The authors would like to acknowledge Arjo Dhidhessa Sugar Factory, local communities. And district experts who assisted the field survey. Also, they would like to thank Ashetu Kejela for preparing the map of the study area and Mokonnen Teshome for Botanical Herbarium, Wollega University and for diet quality analysis.

## Author contributions

**Conceptualization:** Girma Gizachew Tefera, Tadesse Habtamu Tessema, Tibebu Alemu Bekere, Tariku Mekonnen Gutema.

**Data curation:** Girma Gizachew Tefera.

**Formal analysis:** Girma Gizachew Tefera.

**Investigation:** Girma Gizachew Tefera.

**Methodology:** Girma Gizachew Tefera.

**Resources:** Girma Gizachew Tefera.

**Supervision:** Tadesse Habtamu Tessema, Tibebu Alemu Bekere, Tariku Mekonnen Gutema.

**Validation:** Girma Gizachew Tefera, Tadesse Habtamu Tessema, Tibebu Alemu Bekere, Tariku Mekonnen Gutema.

**Visualization:** Girma Gizachew Tefera, Tadesse Habtamu Tessema, Tibebu Alemu Bekere, Tariku Mekonnen Gutema.

**Writing – original draft:** Girma Gizachew Tefera.

**Writing – review & editing:** Girma Gizachew Tefera, Tadesse Habtamu Tessema, Tibebu Alemu Bekere, Tariku Mekonnen Gutema.

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
