## [Decision Letter · Decision Letter 0]

3 Mar 2025

Dear Dr. Tefera,

**Methods and choice of two focal individuals to generalize the behavior are of major concern of common hippos in the DWS. For this, statistical methods need to be clear.**

**Sweeping conclusions on the community aspects, which are not part of the study, may be removed.**

**Classification of behaviors within the context of the study and whether the statistical analysis of the feeding ecology included the faecal samples or only the feeding observations, needs clarification.**

**Certain parts of the manuscript need to be rewritten to address grammar and syntax where the text is difficult to understand.**

**Point wise suggestions and comments by the reviewers need to be addressed for manuscript improvements.**

We look forward to receiving your revised manuscript.

Kind regards,

Munir Ahmad, PhD

Academic Editor

PLOS ONE

**Journal Requirements:**

1. When submitting your revision, we need you to address these additional requirements. Please ensure that your manuscript meets PLOS ONE's style requirements, including those for file naming. The PLOS ONE style templates can be found at https://journals.plos.org/plosone/s/file?id=wjVg/PLOSOne_formatting_sample_main_body.pdf and https://journals.plos.org/plosone/s/file?id=ba62/PLOSOne_formatting_sample_title_authors_affiliations.pdf 2. We note that Figure S1 includes an image of a participant in the study. As per the PLOS ONE policy (http://journals.plos.org/plosone/s/submission-guidelines#loc-human-subjects-research) on papers that include identifying, or potentially identifying, information, the individual(s) or parent(s)/guardian(s) must be informed of the terms of the PLOS open-access (CC-BY) license and provide specific permission for publication of these details under the terms of this license. Please download the Consent Form for Publication in a PLOS Journal (http://journals.plos.org/plosone/s/file?id=8ce6/plos-consent-form-english.pdf). The signed consent form should not be submitted with the manuscript, but should be securely filed in the individual's case notes. Please amend the methods section and ethics statement of the manuscript to explicitly state that the patient/participant has provided consent for publication: “The individual in this manuscript has given written informed consent (as outlined in PLOS consent form) to publish these case details”.  If you are unable to obtain consent from the subject of the photograph, you will need to remove the figure and any other textual identifying information or case descriptions for this individual. 3. When completing the data availability statement of the submission form, you indicated that you will make your data available on acceptance. We strongly recommend all authors decide on a data sharing plan before acceptance, as the process can be lengthy and hold up publication timelines. Please note that, though access restrictions are acceptable now, your entire data will need to be made freely accessible if your manuscript is accepted for publication. This policy applies to all data except where public deposition would breach compliance with the protocol approved by your research ethics board. If you are unable to adhere to our open data policy, please kindly revise your statement to explain your reasoning and we will seek the editor's input on an exemption. Please be assured that, once you have provided your new statement, the assessment of your exemption will not hold up the peer review process. 4. We note that you have referenced Timbuka, C. D. which has currently not yet been accepted for publication. Please remove this from your References and amend this to state in the body of your manuscript: (Timbuka, C. D. [Submitted]) as detailed online in our guide for authorshttp://journals.plos.org/plosone/s/submission-guidelines#loc-reference-style 5. We note that Figure 1 in your submission contain map images which may be copyrighted. All PLOS content is published under the Creative Commons Attribution License (CC BY 4.0), which means that the manuscript, images, and Supporting Information files will be freely available online, and any third party is permitted to access, download, copy, distribute, and use these materials in any way, even commercially, with proper attribution. For these reasons, we cannot publish previously copyrighted maps or satellite images created using proprietary data, such as Google software (Google Maps, Street View, and Earth). For more information, see our copyright guidelines: http://journals.plos.org/plosone/s/licenses-and-copyright. We require you to either present written permission from the copyright holder to publish these figures specifically under the CC BY 4.0 license, or remove the figures from your submission: a. You may seek permission from the original copyright holder of Figure 1 to publish the content specifically under the CC BY 4.0 license.   We recommend that you contact the original copyright holder with the Content Permission Form (http://journals.plos.org/plosone/s/file?id=7c09/content-permission-form.pdf) and the following text:“I request permission for the open-access journal PLOS ONE to publish XXX under the Creative Commons Attribution License (CCAL) CC BY 4.0 (http://creativecommons.org/licenses/by/4.0/). Please be aware that this license allows unrestricted use and distribution, even commercially, by third parties. Please reply and provide explicit written permission to publish XXX under a CC BY license and complete the attached form.” Please upload the completed Content Permission Form or other proof of granted permissions as an "Other" file with your submission. In the figure caption of the copyrighted figure, please include the following text: “Reprinted from [ref] under a CC BY license, with permission from [name of publisher], original copyright [original copyright year].” b. If you are unable to obtain permission from the original copyright holder to publish these figures under the CC BY 4.0 license or if the copyright holder’s requirements are incompatible with the CC BY 4.0 license, please either i) remove the figure or ii) supply a replacement figure that complies with the CC BY 4.0 license. Please check copyright information on all replacement figures and update the figure caption with source information. If applicable, please specify in the figure caption text when a figure is similar but not identical to the original image and is therefore for illustrative purposes only.The following resources for replacing copyrighted map figures may be helpful: USGS National Map Viewer (public domain): http://viewer.nationalmap.gov/viewer/The Gateway to Astronaut Photography of Earth (public domain): http://eol.jsc.nasa.gov/sseop/clickmap/Maps at the CIA (public domain): https://www.cia.gov/library/publications/the-world-factbook/index.html and https://www.cia.gov/library/publications/cia-maps-publications/index.htmlNASA Earth Observatory (public domain): http://earthobservatory.nasa.gov/Landsat: http://landsat.visibleearth.nasa.gov/USGS EROS (Earth Resources Observatory and Science (EROS) Center) (public domain): http://eros.usgs.gov/#Natural Earth (public domain): http://www.naturalearthdata.com/

**Additional Editor Comments:**

Methods and choice of two focal individuals to generalize the behavior are of major concern of common hippos in the DWS. For this, statistical methods need to be clear.

Sweeping conclusions on the community aspects, which are not part of the study, may be removed.

Classification of behaviors within the context of the study and whether the statistical analysis of the feeding ecology included the faecal samples or only the feeding observations, needs clarification.

Certain parts of the manuscript need to be rewritten to address grammar and syntax where the text is difficult to understand.

Point wise suggestions and comments by the reviewers need to be addressed for manuscript improvements.

Reviewers' comments:

Reviewer's Responses to Questions

**Comments to the Author**

1. Is the manuscript technically sound, and do the data support the conclusions?

Reviewer #1: Partly

Reviewer #2: Yes

2. Has the statistical analysis been performed appropriately and rigorously?

Reviewer #1: No

Reviewer #2: Yes

3. Have the authors made all data underlying the findings in their manuscript fully available?

Reviewer #1: Yes

Reviewer #2: Yes

4. Is the manuscript presented in an intelligible fashion and written in standard English?

Reviewer #1: Yes

Reviewer #2: Yes

**Reviewer #1: ** I have made a series of annotations on the pdf attached. My concern is on your methods and choice two focal individuals to generalize the behavior of common hippos in the DWS. Be clear on the statistical methods you used. You made sweeping conclusions on the community aspects which are not part of the study remove them. There is scope for improvement for this manuscript.

**Reviewer #2: ** General comments

This is an interesting and worthwhile study focusing on an often-overlooked megaherbivore, the common hippopotamus. The manuscript provides a comprehensive account of the feeding and activity habits for hippos in the Sanctuary, which could have important future conservation management implications. Overall, the study is well-structured, and the methodology is robust. However, the manuscript would benefit from some further clarification regarding the classification of behaviours within the context of the study and whether the statistical analysis of the feeding ecology included the faecal samples or only the feeding observations. Certain parts of the manuscript also need to be rewritten to address grammar and syntax where the text is difficult to understand and to improve overall readability. There is a repeated paragraph in the discussion that needs to be addressed. Please see further detailed feedback below.

Detailed feedback

Line 69- unclear on what ‘dimensions’ means here, consider rephrasing for clarity

Line 77- add reference here

Line 84- what were the ‘intended purposes’ of the formation of DWS? Also, suggest providing the full name here of DWS as done in abstract

Line 98- remove ‘were’

Line 115- ‘primarily growing’ as in Hyparrhenia spp. are dominant? Or perennially? Maybe clarify

Lines 127-132- the addition of ‘actions’ and ‘occurrences’ is confusing, stick to ‘activities’ and ‘events’ as per the analysis

Lines 131-132- state why this distinction was made and what was considered a short period of time vs a long period of time

Line 129 + 177-178- what types of behaviours were placed in the ‘socialisation’ category? Also specify the behaviours considered ‘mating behaviours’ and how these differ from yawning and barking (as mentioned in the discussion on line 296)

Line 198-199- what about behavioural events as per your definitions?

Line 218-219- suggest rephrasing ‘feeding bites’ for clarity as unclear what this means

Line 239- suggest remove ‘importantly’

Line 238-242- does this statistical test include the faecal samples between the wet and dry seasons as well as the observations? If not, the manuscript would benefit from an additional statistical test of the faecal samples to determine whether the observed differences between the seasons were statistically significant

Lines 256-279- repeated paragraph, suggest remove the second one as lines 277-279 are incomprehensible

Line 283- add references to support these statements, no analysis in this study was undertaken to conclude that these variables significantly altered the activity patterns

Lines 332-336- suggest rephrasing ‘almost all’ since the removal of 12 species out of a total of 34 is a significant amount

Lines 345-346- given the consistency of the results with earlier studies in Ethiopia, how could the outcome of the research contribute to the management of other areas with hippo populations?

**Do you want your identity to be public for this peer review?** For information about this choice, including consent withdrawal, please see our Privacy Policy

Reviewer #1: **Yes: ** Beaven Utete

Reviewer #2: No

---

## [Author Response · Author response to Decision Letter 1]

16 Apr 2025

Date; - March, 28, 2025

To: Munir Ahmad, PhD.

Academic Editor

Journal of PLOS ONE

Subject: Responses to the reviewers’ comments

Dear Dr. Stephanie S. Romanach,

We are very grateful for your and reviewers’ critical and constructive comments regarding our manuscript numbered [PONE-D-24-55622] - [EMID: 8a6d12479cab2714], entitled “Feeding ecology and activity patterns of Hippopotamus amphibious in the Dhidhessa Wildlife Sanctuary, Southwestern Ethiopia”. Please find re-submitted (the revised version) of our manuscript in which we have carefully considered and addressed all comments raised by the referees. The revised version of the manuscript includes many improvements and summarized below. Moreover, the revised manuscript is presented using track-change with red color. We hope our responses are satisfactory and you will find this version suitable for publication in Journal of PLOS ONE. We look forward to hearing from you.

Dear Academic Editor, please note the following points

1. Our responses to the reviewers’ comments are provided

2. The line numbers with red colors (line#) refer to the improved version of the revised manuscript

Best regards,

Corresponding author

Response to Academic editor #1

1. Comment 1(General comment), the reviewers suggested that the manuscript when submitting your revision, we need you to address these additional requirements.

• The manuscript should meet PLOS One's style requirements.

• Data availability by the authors upon reasonable request.

• Figures 1 in your submission contain [map/satellite] images, which may be copyrighted.

Response: Thank you very much for your suggestion. We agree with the comment and we made improvements to the issues mentioned above.

2. Comment 2: Methods and choice of two focal individuals to generalize the behavior are of major concern of common hippos in the DWS. For this, statistical methods need to be clear.

Response: Thank you very much for your suggestion. We agree with the comment and made correction from Line# 126-196. We have re-written or revised the whole of portions the methods.

3. Comment 3: Sweeping conclusions on the community aspects, which are not part of the study, may be removed.

Response: Thank you very much for your comments and suggestions. We have made correction as commented on Line# 402-426.

4. Comment 4: “Classification of behaviors within the context of the study and whether the statistical analysis of the feeding ecology included the faecal samples or only the feeding observations, needs clarification “

Response: Thank you very much for your suggestions. We have made correction as commented on Line# 161-196.

5. Comment 5: Certain parts of the manuscript need to be rewritten to address grammar and syntax where the text is difficult to understand

Response: Thank you very much for your comments and suggestions. We have changed and revised as commented.

6. Comment 6: The reviewers suggested that Please clarify whether written consent for publication has been obtained from the participants shown in [FIGURE S1] in the ‘Consent for publication’ section. The manuscript when submitting your revision, we need you to address these additional requirements.

Response: Thank you very much for your suggestion. Microscopic imaging for fecal analysis was conducted by the corresponding author. Informed consent for publication has been duly obtained from all participants involved in the study.

7. Comment 7: When completing the data availability statement of the submission form, you indicated that you will make your data available on acceptance.

Response: Thank you very much for your comments. We agree with the comment and made correction based as commented.

8. Comment 8: We note that you have referenced Timbuka, C. D. which has currently not yet been accepted for publication. Please remove this from your References and amend this to state in the body of your manuscript: (Timbuka, C. D

Response: Thank you very much for your suggestions. We agree with the comment and made correction by rewriting the words using a track change from Line# 497

9. Comment 9, Line 69: Unclear on what ‘dimensions’ means here, consider rephrasing for clarity

Response: Thank you very much for your suggestions. We have made correction on line #71 – 73. For clarity the dimensions can include things like temperature, food availability, pH, salinity, and other factors that influence an organism's survival and reproduction.

10. Comment 10, Line 84: What were the ‘intended purposes’ of the formation of DWS? Also, suggest providing the full name here of DWS as done in abstract.

Responses: Thank you very much for your comments. We agree with the comment and made correction. Based on the comments and suggestions using a track change from Line# 37,96-100, we have re-written or revised the whole of those portions.

11. Comment 11, 98: remove ‘were’

Response: Thank you very much for your comments/suggestions. We agree with the comment and made correction (line #108).

12. Comment 12, Line 115: ‘primarily growing’ as in Hyparrhenia spp. are dominant? Or perennially? Maybe clarify

Response: Thank you very much for your suggestions. We agree with the comment and made correction from Line# 122-124.

13. Comment 13, line 127-132: “the addition of ‘actions’ and ‘occurrences’ is confusing, stick to ‘activities’ and ‘events’ as per the analysis?

Response: Thank you very much for your comments. We agree with the comment and made correction from Line# 147-150.

14. Comment 14, Line 131-132: State why this distinction was made and what was considered a short period of time vs a long period of time

Response: Thank you very much for your comments and suggestions. We agree with the comment and made correction using a track change from Line# 126-160. ‘’Behavioral activities were defined as sustained actions of relatively longer duration, including: Resting, Standing, Walking, Feeding, Socializing. Behavioral events comprised discrete, short-duration patterns, often lasting seconds, such as: Vocalizations (e.g., barking) Transient body movements (e.g., yawning) “

15. Comment 15, Line 129+177-178: what types of behaviors were placed in the ‘socialisation’ category? Also specify the behaviors considered ‘mating behaviors’ and how these differ from yawning and barking (as mentioned in the discussion on line 296)

Response: Thank you very much for your comments. We agree with the comment and made correction using a track change from Line# 126-160.

16. Comment 16, Line 198-199: What about behavioural events as per your definitions?

Response: Thank you very much for your comments and suggestions. We agree with the comment and made correction using a track change from Line# 141-160

17. Comment 17, Line 218-219: suggest rephrasing ‘feeding bites’ for clarity as unclear what this means

Response: Thank you very much for your suggestions. We agree with the comment and clarified. ‘’To clarify the term "feeding bites," alternative phrasings can be used depending on the focus of observation: "Bites during foraging" explicitly links the action to feeding temporality, while "plant material bites" specifies the target (e.g., grasses, aquatic vegetation) if describing grazing specificity’’.

18. Comment 18, Line 239: Suggest remove ‘importantly’.

Response: Thank you very much for your comments and suggestions. We agree with the comment and made correction using a track change from Line# 276-277.

19. Comment 19, Line 238-242: Does this statistical test include the faecal samples between the wet and dry seasons as well as the observations? If not, the manuscript would benefit from an additional statistical test of the faecal samples to determine whether the observed differences between the seasons were statistically significant.

Response: Thank you very much for your comments. However, we made the comparisons based on the methodologies and objectives rather than the area's size from Line# 285-293.

20. Comment 20, Line 256-279: Repeated paragraph, suggest remove the second one as lines 277-279 are incomprehensible.

Response: Thank you very much for your comment. We agree with the comment and made correction using a track change from Line# 327-345

21. Comment 21, Line 283: Add references to support these statements, no analysis in this study was undertaken to conclude that these variables significantly altered the activity patterns.

Response: Thank you very much for your comment. We agree with the comment and made correction using a track change from Line# 322.

22. Comment 22, Line 332-336: Suggest rephrasing ‘almost all’ since the removal of 12 species out of a total of 34 is a significant amount.

Response: Thank you very much for your comments and suggestions. We agree with the comment and made correction using a track change from Line# 366-373.

23. Comment 23, Line 345-346: Given the consistency of the results with earlier studies in Ethiopia, how could the outcome of the research contribute to the management of other areas with hippo populations

Response: Thank you very much for your suggestions. We have changed as commented, on #Line 198-293.

---

## [Editor Report · Decision Letter 1]

18 May 2025

Feeding ecology and activity patterns of Hippopotamus amphibious in the Dhidhessa Wildlife Sanctuary, Southwestern Ethiopia

PONE-D-24-55622R1

Dear Dr. Tefera,

We’re pleased to inform you that your manuscript has been judged scientifically suitable for publication and will be formally accepted for publication once it meets all outstanding technical requirements.

Kind regards,

Munir Ahmad, PhD

Academic Editor

PLOS ONE
---

## [Editor Report · Acceptance letter]

PONE-D-24-55622R1

PLOS ONE

Dear Dr. Tefera,

I'm pleased to inform you that your manuscript has been deemed suitable for publication in PLOS ONE. Congratulations! Your manuscript is now being handed over to our production team.

Kind regards,

on behalf of

Dr. Munir Ahmad

Academic Editor

PLOS ONE